# Conformal Symplectic and Relativistic Optimization

**Guilherme França**[*]
UC Berkeley
Johns Hopkins

**Jeremias Sulam**
Johns Hopkins

**Daniel P. Robinson**
Lehigh

**René Vidal**
Johns Hopkins

## Abstract

Arguably, the two most popular accelerated or momentum-based optimization methods are Nesterov's accelerated gradient and Polyaks's heavy ball, both corresponding to different discretizations of a particular second order differential equation with a friction term. Such connections with continuous-time dynamical systems have been instrumental in demystifying acceleration phenomena in optimization. Here we study structure-preserving discretizations for a certain class of dissipative (conformal) Hamiltonian systems, allowing us to analyze the symplectic structure of both Nesterov and heavy ball, besides providing several new insights into these methods. Moreover, we propose a new algorithm based on a dissipative relativistic system that normalizes the momentum and may result in more stable/faster optimization. Importantly, such a method generalizes both Nesterov and heavy ball, each being recovered as distinct limiting cases, and has potential advantages at no additional cost.

## 1   Introduction

Gradient based optimization methods are ubiquitous in machine learning since they only require first order information on the objective function. This makes them computationally efficient. However, vanilla gradient descent can be slow. Alternatively, *accelerated gradient methods*, whose construction can be traced back to Polyak [1] and Nesterov [2], became popular due to their ability to achieve best worst-case complexity bounds. The heavy ball method, also known as *classical momentum* (CM) method, is given by

$$v_{k+1} = \mu v_k - \epsilon \nabla f(x_k), \qquad x_{k+1} = x_k + v_{k+1}, \tag{1}$$

where $\mu \in (0, 1)$ is the momentum factor, $\epsilon > 0$ is the learning rate, and $f : \mathbb{R}^n \to \mathbb{R}$ is the function being minimized. Similarly, *Nesterov's accelerated gradient* (NAG) can be found in the form

$$v_{k+1} = \mu v_k - \epsilon \nabla f(x_k + \mu v_k), \qquad x_{k+1} = x_k + v_{k+1}. \tag{2}$$

Both methods have a long history in optimization and machine learning [3]. They are also the basis for the construction of other methods, such as adaptive ones that additionally include some gradient normalization [4–7].

In discrete-time optimization the "acceleration phenomena" are considered counterintuitive. A promising direction has been emerging in connection with continuous-time dynamical systems [8–18] where many of these difficulties disappear or have an intuitive explanation. Since one is free to discretize a continuous system in many different ways, it is only natural to ask which discretization strategies would be most suitable for optimization? Such a question is unlikely to have a simple answer, and may be problem dependent. Unfortunately, typical discretizations are also known to introduce spurious artifacts and do not reproduce the most important properties of the continuous system [19]. Nevertheless, a special class of discretizations in the physics literature

---

[*]To whom correspondence should be addressed: `guifranca@gmail.com`, `guifranca@berkeley.edu`

**Algorithm 1** *Relativistic Gradient Descent* (RGD) for minimizing a smooth function $f(x)$. In practice, we recommend setting $\alpha = 1$ which results in a conformal symplectic method.

---

**Require:** Initial state $(x_0, v_0)$ and parameters $\epsilon > 0$, $\delta > 0$, $\mu \in (0,1)$, $\alpha \in [0,1]$
    **for** $k = 0, 1, \ldots$ **do**
        $x_{k+1/2} \leftarrow x_k + \sqrt{\mu}\, v_k / \sqrt{\mu\delta\|v_k\|^2 + 1}$
        $v_{k+1/2} \leftarrow \sqrt{\mu}v_k - \epsilon \nabla f(x_{k+1/2})$
        $x_{k+1} \leftarrow \alpha x_{k+1/2} + (1-\alpha)x_k + v_{k+1/2}/\sqrt{\delta\|v_{k+1/2}\|^2 + 1}$
        $v_{k+1} \leftarrow \sqrt{\mu}\, v_{k+1/2}$
    **end for**

---

known as *symplectic integrators* [19–22] are to be preferable whenever considering the special class of *conservative Hamiltonian systems*.

More relevant to optimization is a class of *dissipative* systems known as *conformal Hamiltonian systems* [23]. Recently, results from symplectic integrators were extended to this case and such methods are called *conformal symplectic integrators* [18,24]. Conformal symplectic methods tend to have long time stability because the numerical trajectories remain in the same conformal symplectic manifold as the original system [18]. Importantly, these methods do not change the phase portrait of the system, i.e. the stability of critical points is preserved. Although symplectic techniques have had great success in several areas of physics and Monte Carlo methods, only recently they started to be considered in optimization [14,18] and are still mostly unexplored in this context. Very recently a great progress has been made [18] by showing that such an approach is able to preserve the continuous-time rates of convergence up to a controlled error [18].

In this paper, we *relate conformal symplectic integrators to optimization* and provide important insights into CM (1) and NAG (2). We prove that CM is a first order accurate conformal symplectic integrator. On the other hand, we show that NAG is also first order accurate, but not conformal symplectic since it introduces some spurious dissipation (or excitation). However, it does so in an interesting way that depends on the Hessian $\nabla^2 f$; the symplectic form contracts in a Hessian dependent manner and so do phase space volumes. This is an effect of higher order but can influence the behaviour of the algorithm. We also derive *modified equations* and *shadow Hamiltonians* for both CM and NAG. Moreover, we indicate a tradeoff between stability, symplecticness, and such an spurious contraction, indicating advantages in structure-preserving discretizations for optimization.

Optimization can be challenging in a landscape with large gradients, e.g. for a function with fast growing tails. The only way to control divergences in methods such as (1) and (2) is to make the step size very small, but then the algorithm becomes slow. One approach to this issue is to introduce a suitable normalization of the gradient. Here we propose an alternative approach motivated by *special relativity* in physics. The reason is that in special relativity there is a limiting speed, i.e. the speed of light. Thus, by discretizing a *dissipative relativistic system*, we obtain an algorithm that incorporates this effect and may result in more stable optimization in settings with large gradients. Specifically, we introduce Algorithm 1. Besides the momentum factor $\mu$ and the learning rate $\epsilon$—also present in (1) and (2)—the above RGD method has the additional parameters $\delta \geq 0$ and $0 \leq \alpha \leq 1$ which brings some interesting properties:

- When $\delta = 0$ and $\alpha = 0$, RGD recovers NAG (2). When $\delta = 0$ and $\alpha = 1$, RGD becomes a second order accurate version of CM (1), which has a close behavior but an improved stability. Thus, RGD can interpolate between these two methods. Moreover, RGD has the same computational cost as CM or NAG. These facts imply that RGD is at least as efficient as CM and NAG if appropriately tuned.

- Let $y_k \equiv \alpha x_{k+1/2} + (1-\alpha)x_k$. The last update in Algorithm 1 implies $\|x_{k+1} - y_k\| \leq 1/\delta$. Thus, with $\delta > 0$, RGD is globally bounded regardless how large $\|\nabla f\|$ might be; this is in contrast with CM and NAG where $\delta = 0$, i.e. $\|x_{k+1} - y_k\| \leq \infty$. The square root factor in Algorithm 1 has a "relativistic origin" and its strength is controlled by $\delta$. For this reason, RGD may be more stable compared to CM or NAG, potentially preventing divergences in settings of large gradients; see Fig. 3 in Appendix C and the plots in Appendix D.

- As we will show, $\alpha = 1$ implies that RGD is *conformal symplectic*, whereas $\alpha = 0$ implies a spurious Hessian driven damping similarly found in NAG. Thus, RGD has the flexibility

of being "dissipative-preserving" or introducing some extra "spurious contraction." However, based on theoretical arguments and empirical evidence, we advocate for $\alpha = 1$.[2]

Let us mention a few related works. Applications of symplectic integrators in optimization was first considered in [14]—although this is different than the conformal symplectic case explored here. Recently, the benefits of symplectic methods in optimization started to be indicated [25]. Actually, even more recently, a generalization of symplectic integrators to a general class of dissipative Hamiltonian systems was proposed [18], with theoretical results ensuring that such discretizations are "rate-matching" up to a negligible error; this construction is general and contains the conformal case considered here as a particular case. Relativistic systems are obviously an elementary topic in physics but—with some modifications—the relativistic kinetic energy was considered in Monte Carlo methods [26, 27] and also briefly in [28]. Finally, we stress that Algorithm 1 is a completely new method in the literature, generalizing perhaps the two most popular existing accelerated methods, namely CM and NAG, and also has the ability to be conformal symplectic besides being adaptive in the momentum which may help controlling divergences. We also provide several new insights into CM and NAG in Appendices B and C which may be of independent interest.

## 2   Conformal Hamiltonian Systems

We start by introducing the basics of conformal Hamiltonian systems and focus on their intrinsic symplectic geometry; we refer to [23] for details. The state of the system is described by a point on phase space, $(x, p) \in \mathbb{R}^{2n}$, where $x = x(t)$ is the generalized coordinates and $p = p(t)$ its conjugate momentum, with $t \in \mathbb{R}$ being the time. The system is completely specified by a Hamiltonian function $H : \mathbb{R}^{2n} \to \mathbb{R}$ and required to obey a modified form of Hamilton's equations:

$$\dot{x} = \nabla_p H(x, p), \qquad \dot{p} = -\nabla_x H(x, p) - \gamma p. \tag{3}$$

Here $\dot{x} \equiv \frac{dx}{dt}, \dot{p} \equiv \frac{dp}{dt}$, and $\gamma > 0$ is a damping constant. A classical example is given by

$$H(x, p) = \frac{\|p\|^2}{2m} + f(x) \tag{4}$$

where $m > 0$ is the mass of a particle subject to a potential $f$. The Hamiltonian is the energy of the system and upon taking its time derivative one finds $\dot{H} = -\gamma \|p\|^2 \leq 0$. Thus $H$ is a Lyapunov function and all orbits tend to critical points, which in this case must satisfy $\nabla f(x) = 0$ and $p = 0$. This implies that the system is stable on isolated minimizers of $f$.[3]

Define

$$z \equiv \begin{bmatrix} x \\ p \end{bmatrix}, \qquad \Omega \equiv \begin{bmatrix} 0 & I \\ -I & 0 \end{bmatrix}, \qquad D \equiv \begin{bmatrix} 0 & 0 \\ 0 & I \end{bmatrix}, \tag{5}$$

where $I$ is the $n \times n$ identity matrix, to write the equations of motion (3) concisely as[4]

$$\dot{z} = \underbrace{\Omega \nabla H(z)}_{C(z)} - \underbrace{\gamma D z}_{D(z)}. \tag{6}$$

Note that $\Omega \Omega^T = \Omega^T \Omega = I$ and $\Omega^2 = -I$, so that $\Omega$ is real, orthogonal and antisymmetric. Let $\xi, \eta \in \mathbb{R}^{2n}$ and define the *symplectic 2-form* $\omega(\xi, \eta) \equiv \xi^T \Omega \eta$. It is convenient to use the wedge product representation of this 2-form, namely[5]

$$\omega(\xi, \eta) = (dx \wedge dp)(\xi, \eta). \tag{7}$$

We denote $\omega_t \equiv dx(t) \wedge dp(t)$. The equations of motion define a flow $\Phi_t : \mathbb{R}^{2n} \to \mathbb{R}^{2n}$, i.e. $\Phi_t(z_0) \equiv z(t)$ where $z(0) \equiv z_0$. Let $J_t(z)$ denote the Jacobian of $\Phi_t(z)$. From (6) it is not hard to show that (see e.g. [23])

$$J_t^T \Omega J_t = e^{-\gamma t}\Omega \quad \implies \quad \omega_t = e^{-\gamma t}\omega_0. \tag{8}$$

Therefore, a conformal Hamiltonian flow $\Phi_t$ *contracts the symplectic form exponentially* with respect to the damping coefficient $\gamma$. It follows from (8) that volumes on phase space shrink as $\mathrm{vol}(\Phi_t(\mathcal{R})) = \int_{\mathcal{R}} |\det J_t(z)| dz = e^{-n\gamma t}\mathrm{vol}(\mathcal{R})$ where $\mathcal{R} \subset \mathbb{R}^{2n}$. This contraction is stronger as dimension increases. The conservative case is recovered with $\gamma = 0$ above; in this case, the symplectic structure is preserved and volumes remain invariant (Liouville's theorem). A known and interesting property of conformal Hamiltonian systems is that their Lyapunov exponents sum up in pairs to $\gamma$ [31]. This imposes constraints on the admissible dynamics and controls the phase portrait near critical points. For other properties of attractor sets we refer to [32]. Finally, conformal symplectic transformations can be composed and form the so-called conformal group.

## 3  Conformal Symplectic Optimization

Consider (6) where we associate flows $\Phi_t^C$ and $\Phi_t^D$ to the respective vector fields $C(z)$ and $D(z)$. Conformal symplectic integrators can be constructed as *splitting methods* that approximate the true flow $\Phi_t$ by composing the individual flows $\Phi_t^C$ and $\Phi_t^D$. Our procedure to obtain a numerical map $\Psi_h$, with step size $h > 0$, is to first obtain a numerical approximation to the conservative part of the system, $\dot{z} = \Omega\nabla H(z)$. This yields a numerical map $\Psi_h^C$ that approximates $\Phi_h^C$ for small intervals of time $[t, t + h]$. One can choose any standard *symplectic integrator* for this task. Let us pick the simplest, i.e. the symplectic Euler method [30, pp. 189]. We thus have $\Psi_h^C : (x, p) \mapsto (X, P)$ where

$$X = x + h\nabla_p H(x, P), \qquad P = p - h\nabla_x H(x, P). \tag{9}$$

Now the dissipative part of the system, $\dot{z} = -\gamma D z$, can be integrated exactly. Indeed, $\dot{x} = 0$ and $\dot{p} = -\gamma p$, thus $\Psi_h^D : (x, p) = (x, e^{-\gamma h}p)$. With $\Psi_h \equiv \Psi_h^C \circ \Psi_h^D$ we obtain $\Psi_h : (x, p) \mapsto (X, P)$ as

$$P = e^{-\gamma h}p - h\nabla_x H(x, P), \qquad X = x + h\nabla_p H(x, P). \tag{10}$$

This is nothing but a *dissipative version* of the symplectic Euler method. Similarly, if we choose the leapfrog method [30, pp. 190] for $\Psi_h^C$ and consider $\Psi_h \equiv \Psi_{h/2}^D \circ \Psi_h^C \circ \Psi_{h/2}^D$ we obtain

$$\tilde{X} = x + \tfrac{h}{2}\nabla_p H(\tilde{X}, e^{-\gamma h/2}p), \tag{11a}$$

$$\tilde{P} = e^{-\gamma h/2}p - \tfrac{h}{2}\big(\nabla_x H(\tilde{X}, e^{-\gamma h/2}p) + \nabla_x H(\tilde{X}, \tilde{P})\big), \tag{11b}$$

$$X = \tilde{X} + \tfrac{h}{2}\nabla_p H(\tilde{X}, \tilde{P}), \tag{11c}$$

$$P = e^{-\gamma h/2}\tilde{P}. \tag{11d}$$

This is a *dissipative* version of the leapfrog, which is recovered when $\gamma = 0$. Note that in general (10) is implicit in $P$, and (11) is implicit in $\tilde{X}$ and $P$. However, both will become explicit for separable Hamiltonians, $H = T(p) + f(x)$, and in this case they are extremely efficient. Note also that (10) and (11) are completely general, i.e. by choosing a suitable Hamiltonian $H$ one can obtain several possible optimization algorithms from these integrators. Next, we show important properties of these integrators. (Below we denote $t_k = kh$ for $k = 0, 1, \ldots, z_k \equiv z(t_k)$, etc.)

**Definition 1.** *A numerical map $\Psi_h$ is said to be of order $r \geq 1$ if $\|\Psi_h(z) - \Phi_h(z)\| = O(h^{r+1})$ for any $z \in \mathbb{R}^{2n}$. (Recall that $h > 0$ is the step size and $\Phi_h$ is the true flow.)*

**Definition 2.** *A numerical map $\Psi_h$ is said to be conformal symplectic if $z_{k+1} = \Psi_h(z_k)$ is conformal symplectic, i.e. $\omega_{k+1} = e^{-\gamma h}\omega_k$, whenever $\hat{\Phi}_h$ is applied to a smooth Hamiltonian. Iterating such a map yields $\omega_k = e^{-\gamma t_k}\omega_0$ so that (8) is preserved.*

**Theorem 3.** *Both methods (10) and (11) are conformal symplectic.*

*Proof.* Note that in both cases $\Psi_h^C$ is a symplectic integrator, i.e. its Jacobian $J_h^C$ obeys $(J_h^C)^T \Omega J_h^C = \Omega$—see (8) with $\gamma = 0$. Now the map $\Psi_h^D$ defined above is conformal symplectic, i.e. one can verify that its Jacobian $J_h^D$ obeys $(J_h^D)^T \Omega J_h^D = e^{-\gamma h}\Omega$. Hence, any composition of these maps will be conformal symplectic. For instance,

$$(J_h^C J_h^D)^T \Omega(J_h^C J_h^D) = (J_h^D)^T (J_h^C)^T \Omega J_h^C J_h^D = (J_h^D)^T \Omega J_h^D = e^{-\gamma h}\Omega. \tag{12}$$

The same would be true for any type of composition whose overall time step add up to $h$. $\qquad\square$

**Theorem 4.** *The numerical scheme* (10) *is of order* $r = 1$*, while* (11) *is of order* $r = 2$*.*

*Proof.* The proof simply involves manipulating Taylor expansions for the numerical method and the continuous system over a time interval of $h$; this is presented in Appendix A. $\qquad\square$

We mention that one can construct higher order integrators by following the above approach, however these would be more expensive, involving more gradient computations per iteration. In practice, methods of order $r = 2$ tend to have the best cost benefit.

## 4 Symplectic Structure of Heavy Ball and Nesterov

Consider the classical Hamiltonian (4) and replace into (10) to obtain

$$p_{k+1} = e^{-\gamma h} p_k - h\nabla f(x_k), \qquad x_{k+1} = x_k + \tfrac{h}{m} p_{k+1}, \tag{13}$$

where we now make the iteration number $k = 0, 1, \ldots$ explicit for convenience of the reader in relating to optimization methods. Introducing a change of variables,

$$v_k \equiv \tfrac{h}{m} p_k, \qquad \epsilon \equiv \tfrac{h^2}{m}, \qquad \mu \equiv e^{-\gamma h}, \tag{14}$$

we see that (13) is precisely the well-known CM method (1). Therefore, CM is nothing but a dissipative version of the symplectic Euler method. Thanks to Theorems 3 and 4 we have:

**Corollary 5.** *The classical momentum* (1)*, or heavy ball, is a conformal symplectic integrator for the Hamiltonian system* (4)*. Moreover, it is an integrator of order* $r = 1$*.*

Consider again the Hamiltonian (4) but replaced into (11). Let us also replace the last update (11d), i.e. from a previous iteration, into the first update (11a).[6] We thus obtain

$$x_{k+1/2} = x_k + \tfrac{h}{2m} e^{-\gamma h} p_k, \quad p_{k+1} = e^{-\gamma h} p_k - h\nabla f(x_{k+1/2}), \quad x_{k+1} = x_{k+1/2} + \tfrac{h}{2m} p_{k+1}. \tag{15}$$

Define

$$v_k \equiv \tfrac{h}{2m} p_k, \qquad \epsilon \equiv \tfrac{h^2}{2m}, \qquad \mu \equiv e^{-\gamma h}. \tag{16}$$

Then (15) can be written as

$$x_{k+1/2} = x_k + \mu v_k, \qquad v_{k+1} = \mu v_k - \epsilon \nabla f(x_{k+1/2}), \qquad x_{k+1} = x_{k+1/2} + v_{k+1}. \tag{17}$$

The reader can immediately recognize the close similarity with NAG (2); this would be exactly NAG if we replace $x_{k+1/2} \to x_k$ in the third update above. As we will show next, this small difference has actually profound consequences. Intuitively, by "rolling this last update backwards" one introduces a spurious friction into the method, as we will show through a symplectic perspective (Theorem 6 below). The method (15) is actually a second order accurate version of (13). In order to analyze the symplectic structure one must work on the phase space $(x, p)$. The true phase space equivalent to NAG is given by

$$x_{k+1/2} = x_k + \tfrac{h}{m} e^{-\gamma h} p_k, \quad p_{k+1} = e^{-\gamma h} p_k - h\nabla f(x_{k+1/2}), \quad x_{k+1} = x_k + \tfrac{h}{m} p_{k+1}, \tag{18}$$

which is completely equivalent to (2) under the correspondence (14). We thus have the following.

**Theorem 6.** *Nesterov's accelerated gradient* (2)*, or equivalently* (18)*, is an integrator of order* $r = 1$ *to the Hamiltonian system* (4)*. This method is not conformal symplectic but rather contracts the symplectic form as*

$$\omega_{k+1} = e^{-\gamma h} \left[ I - \tfrac{h^2}{m} \nabla^2 f(x_k) \right] \omega_k + O(h^3). \tag{19}$$

*Proof.* The details are presented in Appendix B, though the argument is simple. First, compare Taylor expansions of (18) and $(x(t+h), p(t+h))$. Second, use the variational form of (18), replace into $dx_{k+1} \wedge dp_{k+1}$, and then use basic properties of the wedge product. $\qquad\square$

### 4.1 Alternative Form

It is perhaps more common to find Nesterov's method in the following form:

$$x_{k+1} = y_k - \epsilon \nabla f(y_k), \qquad y_{k+1} = x_{k+1} + \mu_{k+1}(x_{k+1} - x_k), \qquad (20)$$

where $\mu_{k+1} = k/(k+3)$. This is equivalent to (2), as can be seen by introducing the variable $v_k \equiv x_k - x_{k-1}$ and writing the updates in terms of $x$ and $v$. When $\mu_k$ is constant, Theorem 6 shows that the method is not conformal symplectic. When $\mu_k = k/(k+3)$, the differential equation associated to (20) is equivalent to (3)/(4) with $\gamma = 3/t$. It is possible to generalize the above results for time dependent cases [18]. Therefore, also in this case, NAG does not preserve the symplectic structure; we note that (19) still holds with $e^{-\gamma h} \to e^{-3\log(1+h/t_k)}$ where $t_k = hk$.

### 4.2 Preserving Stability and Continuous Rates

An important question is whether being symplectic is beneficial or not for optimization. Very recently, it has been shown [18] that symplectic discretizations of dissipative systems may indeed preserve continuous-time rates of convergence when $f$ is smooth and the system is appropriately dampened (choice of $\gamma$); the continuous-time rates can be obtained via Lyapunov analysis. Thus, assuming that we have a suitable conformal Hamiltonian system, conformal symplectic integrators provide a principled approach to construct optimization algorithms which are guaranteed to respect the main properties of the system, such as stability of critical points and convergence rates. Furthermore, we claim that there is a delicate tradeoff where being conformal symplectic is related to an improved stability, in the sense that the method can operate with larger step sizes, while the spurious dissipation introduced by NAG (Theorem 6) may improve the convergence rate slightly, since it introduces more contraction, but at the cost of making the method less stable. Due to the lack of space, we defer these details to Appendix C. In Appendix B we also provide important additional insights into CM and NAG, such as their modified or perturbed equations and their *shadow Hamiltonians*, which describe these methods to a higher degree of resolution.

## 5 Optimization from a Dissipative Relativistic System

Let us briefly mention some simple but fundamental concepts to motivate our approach. The previous algorithms are based on (4) which leads to a classical Newtonian system where time is just a parameter, independent of the Euclidean space where the trajectories live. This implies that there is no restriction on the speed, $\|v\| = \|dx/dt\|$, that a particle can attain. This translates to a discrete-time algorithm, such as (13), where large gradients $\nabla f$ give rise to a large momenta $p$, implying that the position updates for $x$ can diverge. On the other hand, in special relativity, space and time form a unified geometric entity, the $(n+1)$-dimensional Minkowski spacetime with coordinates $X = (ct; x)$, where $c$ denotes the speed of light. An infinitesimal distance on this manifold is given by $ds^2 = -(cdt)^2 + \|dx\|^2$. Null geodesics correspond to $ds^2 = 0$, implying $\|v\|^2 = \|dx/dt\|^2 = c^2$, i.e. no particle can travel faster than $c$. This imposes constraints on the geometry where trajectories take place—it is actually a hyperbolic geometry. With that being said, the idea is that by discretizing a relativistic system we can incorporate these features into an optimization algorithm which may bring benefits such as an improved stability.

A relativistic particle subject to a potential $f$ is described by the following Hamiltonian:

$$H(x,p) = c\sqrt{\|p\|^2 + m^2c^2} + f(x). \qquad (21)$$

In the classical limit, $\|p\| \ll mc$, one obtains $H = mc^2 + \|p\|^2/(2m) + f(x) + O(1/c^2)$, recovering (4) up to the constant $E_0 = mc^2$, which has no effect in deriving the equations of motion. Replacing (21) into (3) we thus obtain a *dissipative relativistic system*:

$$\dot{x} = \frac{cp}{\sqrt{\|p\|^2 + m^2c^2}}, \qquad \dot{p} = -\nabla f - \gamma p. \qquad (22)$$

Importantly, in (22) the momentum is normalized by the $\sqrt{\cdot}$ factor, so $\dot{x}$ remains bounded even if $p$ was to go unbounded. Now, replacing (21) into the first order accurate conformal symplectic integrator (10), we readily obtain

$$p_{k+1} = e^{-\gamma h} p_k - h\nabla f(q_k), \qquad x_{k+1} = x_k + \frac{hcp_{k+1}}{\sqrt{\|p_{k+1}\|^2 + m^2c^2}}. \qquad (23)$$

When $c \to \infty$ the above updates recover CM (13). Thus, this method is a relativistic generalization of CM or heavy ball. Moreover, the method (23) is a first order conformal symplectic integrator by construction (see Theorems 3 and 4).

One can replace the Hamiltonian (21) into (11) to obtain a second order version of (23). However, motivated by the close connection between NAG and (15)—recall the comments following (17) about NAG "rolling back" the last update—let us additionally introduce a convex combination, $\alpha x_{k+1/2} + (1 - \alpha)x_k$ where $0 \le \alpha \le 1$, between the initial and midpoint of the method. In this manner, we can interpolate between a conformal symplectic regime and a spurious Hessian damping regime (recall Theorem 6). Therefore, we obtain the following integrator:

$$x_{k+1/2} = x_k + (hc/2)e^{-\gamma h/2}p_k \Big/ \sqrt{e^{-\gamma h}\|p_k\|^2 + m^2c^2}, \tag{24a}$$

$$p_{k+1/2} = e^{-\gamma h/2}p_k - h\nabla f(x_{k+1/2}), \tag{24b}$$

$$x_{k+1} = \alpha x_{k+1/2} + (1 - \alpha)x_k + (hc/2)p_{k+1/2}\Big/\sqrt{\|p_{k+1/2}\|^2 + m^2c^2}, \tag{24c}$$

$$p_{k+1} = e^{-\gamma h/2}p_{k+1/2}. \tag{24d}$$

We call this method *Relativistic Gradient Descent* (RGD). By introducing

$$v_k \equiv \frac{h}{2m}p_k, \qquad \epsilon \equiv \frac{h^2}{2m}, \qquad \mu \equiv e^{-\gamma h}, \qquad \delta \equiv 4/(ch)^2, \tag{25}$$

the updates (24) assume the equivalent form stated in Algorithm 1 in the introduction.

RGD (24) (resp. Algorithm 1) has several interesting limits, recovering the behaviour of known algorithms as particular cases. For instance, when $c \to \infty$ (resp. $\delta \to 0$) it reduces to an interpolation between CM (13) (resp. (1)) and NAG (18) (resp. (2)). If we additionally set $\alpha = 0$ it becomes precisely NAG, whether when $\alpha = 1$ it becomes a second order version (in terms of accuracy) of CM.[7] When $\alpha = 1$, and arbitrary $c$ (or $\delta$), RGD is a conformal symplectic integrator thanks to Theorems 3. Recall also that Theorem 4 implies that RGD is a second order accurate integrator. When $\alpha = 0$, and arbitrary $c$ (or $\delta$), RGD is no longer conformal symplectic and introduces a Hessian driven damping in the spirit of NAG. Finally, the parameter $c$ (or $\delta$) controls the strength of the normalization term in the position updates of (24) (or Algorithm 1), which can help preventing divergences when navigating through a rough landscape with large gradients, or fast growing tails. Indeed, note that $\|x_{k+1} - \alpha x_{k+1/2} - (1 - \alpha)x_k\| \le 1/\delta$ is always bounded for $\delta > 0$; this becomes unbounded when $\delta \to 0$, i.e. in the classical limit of CM and NAG.

In short, RGD is a novel algorithm with quite some flexibility and unique features, generalizing perhaps the two most important accelerated gradient based methods in the literature, which can be recovered as limiting cases. Next, we illustrate numerically through simple yet insightful examples that RGD can be more stable and faster than CM and NAG.

## 6 Numerical Experiments

Let us compare RGD (Algorithm 1) against NAG (2) and CM (1) on some test problems. We stress that all hyperparameters of each of these methods were systematically optimized through Bayesian optimization [33] (the default implementation uses a Tree of Parzen estimators). This yields *optimal* and *unbiased* parameters automatically. Moreover, by checking the distribution of these hyperparameters during the tuning process we can get intuition on the sensitivity of each method. Thus, for each algorithm, we show its convergence rate in Fig. 1 when the best hyperparameters were used. In addition, in Fig. 2 we show the distribution of hyperparameters during the Bayesian optimization step—the parameters are indicated and color lines follow Fig. 1. Such values are obtained only when the respective algorithm was able to converge. We note that usually CM and NAG diverged more often than RGD which seemed more robust to parameter choice. Below we describe some of the optimization problems where such algorithms were tested over. In Appendix D we provide several additional experiments illustrating the benefits of RGD. The actual code related to our implementation is extremely simple and can be found at [34].

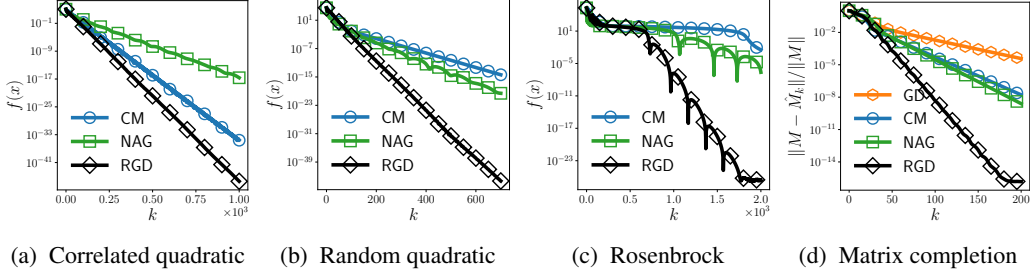

(a) Correlated quadratic    (b) Random quadratic    (c) Rosenbrock    (d) Matrix completion

Figure 1: Convergence rate showing improved performance of RGD (Algorithm 1); see text.

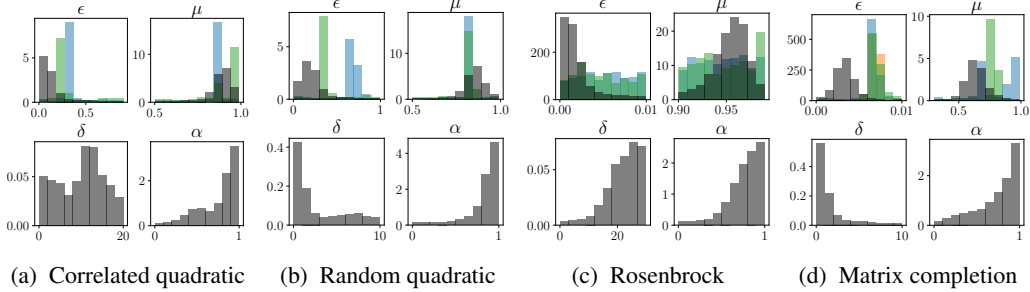

(a) Correlated quadratic    (b) Random quadratic    (c) Rosenbrock    (d) Matrix completion

Figure 2: Histograms of hyperparameter tuning by Bayesian optimization. Tendency towards $\alpha \approx 1$ indicates benefits of being symplectic, while $\alpha \approx 0$ of being extra damped as in NAG. Tendency towards $\delta > 0$ indicates benefits of relativistic normalization. (Color line follows Fig. 1.)

**Correlated quadratic**   Consider $f(x) = (1/2)x^T Q x$ where $Q_{ij} = \rho^{|i-j|}$, $\rho = 0.95$, and $Q$ has size $50 \times 50$—this function was also used in [14]. We initialize the position at random, $x_{0,i} \sim \mathcal{N}(0, 10)$, and the velocity as $v_0 = 0$. The convergence results are shown in Fig. 1a. The distribution of parameters during tuning are in Fig. 2a, showing that $\alpha \to 1$ is preferable. This gives evidence for an advantage in being conformal symplectic. Note also that $\delta > 0$, thus "relativistic effects" played a role in improving convergence.

**Random quadratic**   Consider $f(q) = (1/2)x^T Q x$ where $Q$ is a $500 \times 500$ positive definite random matrix with eigenvalues uniformly distributed in $[10^{-3}, 10]$. Convergence rates are in Fig. 1b with the histograms of parameter search in Fig. 2b. Again, there is a preference towards $\alpha \to 1$, evidencing benefits in being conformal symplectic.

**Rosenbrock**   For a challenging problem in higher dimensions, consider the nonconvex Rosenbrock function $f(x) \equiv \sum_{i=1}^{n-1} \left( 100(x_{i+1} - x_i^2)^2 + (1 - x_i)^2 \right)$ with $n = 100$ [35,36]; this case was already studied in detail [37]. Its landscape is quite involved, e.g. there are two minimizers, one global at $x^\star = (1, \ldots, 1)^T$ with $f(x^\star) = 0$ and one local near $x \approx (-1, 1, \ldots, 1)^T$ with $f \approx 3.99$. There are also—exponentially—many saddle points [37], however only two of these are actually hard to escape. These four stationary points account for $99.9\%$ of the solutions found by Newton's method [37]. We note that both minimizers lie on a flat, deep, and narrow valley, making optimization challenging. In Fig. 1c we have the convergence of each method initialized at $x_{0,i} = \pm 2$ for $i$ odd/even. Fig. 2c shows histograms for parameter selection. Again, we see the favorable symplectic tendency, $\alpha \to 1$. Here relativistic effects, $\delta \neq 0$, played a predominant role in the improved convergence of RGD.

**Matrix completion**   Consider an $n \times n$ matrix $M$ of rank $r \ll n$ with observed entries in the support $(i, j) \in \Omega$, where $P_\Omega(M)_{ij} = M_{ij}$ if $(i, j) \in \Omega$ and $P_\Omega(M)_{ij} = 0$ projects onto this support. The goal is to recover $M$ from the knowledge of $P_\Omega(M)$. We assume that the rank $r$ is known. In this case, if the number of observed entries is $O(rn)$ it is possible to recover $M$ with high probability [38]. We do this by solving the nonconvex problem $\min_{U,V} \|P_\Omega(M - UV^T)\|_F^2$, where $U, V \in \mathbb{R}^{n \times r}$, by *alternating minimization*: for each iteration we apply the previous algorithms first on $U$ with $V$ held fixed, followed by similar updates for $V$ with the new $U$ fixed. This is a know

technique for gradient descent (GD), which we additionally include as a baseline. We generate $M = RS^T$ where $R, S \in \mathbb{R}n \times r$ have i.i.d. entries from the normal distribution $\mathcal{N}(1, 2)$. We initialize $U$ and $V$ sampled from the standard normal. The support is chosen uniformly at random with sampling ratio $s = 0.3$, yielding $p = sn^2$ observed entries. We set $n = 100$ and $r = 5$. This gives a number of effective degrees of freedom $d = r(2n - r)$ and the "hardness" of the problem can be quantified via $d/p \approx 0.325$. Fig. 1d shows the convergence rate, and Fig. 2d the parameter search.

## 7    Discussion and Outlook

This paper introduces a new perspective on a recent line of research connecting accelerated optimization methods to continuous dynamical systems. We brought *conformal symplectic* techniques for *dissipative systems* into this context, besides proposing a new method called *Relativistic Gradient Descent* (RGD) which is based on a dissipative relativistic system; see Algorithm 1. RGD generalizes both the well-know classical momentum (CM), given by (1), as well as Nesterov's accelerated gradient (NAG), given by (2); each of these methods are recovered as particular cases from RGD which has no additional computational cost compared to CM and NAG. Moreover, RGD has more flexibility, can interpolate between a conformal symplectic behaviour or introduce some Hessian dependent damping in the spirit of NAG, and has potential to control instabilities due to large gradients by normalizing the momentum. In our experiments, RGD significantly outperformed CM and NAG, specially in settings with large gradients or functions with a fast growth; besides Section 6 we report additional numerical results in Appendix D.

We also elucidated what is the symplectic structure behind CM and NAG. We found that the former turns out to be a conformal symplectic integrator (Corollary 5), thus being "dissipative-preserving," while the latter introduces a spurious contraction of the symplectic form by a Hessian driven damping (Theorem 6). This is an effect of second order in the step size but may affect convergence and stability. We pointed out that there is a tradeoff between this extra contraction and the stability of a conformal symplectic method; these ideas are explored in more detail in the Appendix C. We also derived modified or perturbed equations for CM and NAG, describing these methods to a higher degree of resolution; this analysis provides several new insights into these methods that were not previously considered in the literature.

On a higher level, this paper shows how structure-preserving discretizations of classical dissipative systems can be useful for studying existing optimization algorithms in machine learning, as well as introduce new methods inspired by real physical systems. A thorough justification for the use of structure-preserving—or "dissipative symplectic"—discretizations in this context was recently provided in [18] under great generality.

A more refined analysis of RGD is certainly an interesting future problem, though considerably challenging due to the nonlinearity introduced by the $\sqrt{1 + \delta\|v\|^2}$ term in the updates of Algorithm 1. To give an example, even if one assumes a simple quadratic function $f(x) = (\lambda/2)x^2$, the differential equation (22) is nonlinear and does not admit a closed form solution, contrary to the differential equation associated to CM and NAG which is linear and can be readily integrated. Thus, even in continuous-time, the analysis for RGD is likely to be involved. Finally, it would be interesting to consider RGD in a stochastic setting, namely investigate its diffusive properties in a random media, which may bring benefits to nonconvex optimization and sampling.

## Broader Impact

The techniques introduced in this paper significantly broaden the applicability of existing results in dynamical systems theory, classical Lagrangian/Hamiltonian mechanics, and numerical analysis of differential equation to optimization, which have widespread applications in several areas of statistical machine learning. This work also lead to cross-fertilization between dynamical systems theory, physics, machine learning and optimization, which can impact emerging work at the intersection of learning and control. We do not see any disadvantages or implications due to failure of this research at this point.

**Acknowledgments**

This work was supported by grants ARO MURI W911NF-17-1-0304, NSF 2031985, and NSF 1934931.

## Footnotes

[2]The only reason for introducing this extra parameter, $0 \leq \alpha \leq 1$, into Algorithm 1 is to actually let the experiments decide whether $\alpha = 1$ (symplectic) or $\alpha < 1$ (non-symplectic) is desirable or not.

[3]This can actually be generalized for any $H$ that is strongly convex on $p$ with the minimum at $p = 0$.

[4]$C(z)$ and $D(z)$ will be used later on and stand for "conservative" and "dissipative" parts, respectively.

[5]It is not strictly necessary to be familiar with differential forms and exterior calculus to understand this paper. For the current purposes, it is enough to recall that the wedge product is bilinear and antisymmetric, i.e. $dx \wedge (a\,dy + b\,dz) = a\,dx \wedge dy + b\,dx \wedge dz$ and $dx \wedge dy = -dy \wedge dx$ for scalars $a$ and $b$ and 1-forms $dx$, $dy$, $dz$ (think about this as vector differentials); we refer to [29] and [30] for more details if necessary.

[6]Note that it is valid to replace successive updates without changing the algorithm.

[7]The dynamics of both CM and this second order version is pretty close, and if anything the latter is even more stable than the former (see Appendix C).

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
