[Supplementary Material]

*Supplemental Material*

# Conformal Symplectic and Relativistic Optimization

**Guilherme França**[*]      **Jeremias Sulam**      **Daniel P. Robinson**      **René Vidal**

In this supplemental material we provide the remaining mathematical proofs and several additional details not discussed in the main part of the paper. Moreover, we complement the numerical section with many other examples.

## A  Order of Accuracy of the General Integrators

It is known that a composition of the type $\Psi_h^A \circ \Psi_h^B$, where $A$ and $B$ represents the components of distinct vector fields, leads to an integrator of order $r = 1$, whereas a composition in the form $\Psi_{h/2}^A \circ \Psi_h^B \circ \Psi_{h/2}^A$ leads to an integrator of order $r = 2$ [30]—the latter is known as Strang splitting. However, here we provide an explicit and direct proof of these facts for the generic integrators (10) and (11), respectively.

*Proof of Theorem 4.*  From the equations of motion (3) and Taylor expansions:

$$
\begin{aligned}
x(t_k + h) &= x + h\dot{x} + \tfrac{h^2}{2}\ddot{x} + O(h^3) \\
&= x + h\nabla_p H + \tfrac{h^2}{2}\left(\nabla_{xp}^2 H\dot{x} + \nabla_{pp}^2 H\dot{p}\right) + O(h^3) \\
&= x + h\nabla_p H + \tfrac{h^2}{2}\nabla_{xp}^2 H\nabla_p H - \tfrac{h^2}{2}\nabla_{xp}^2 H\nabla_x H - \tfrac{h^2}{2}\gamma\nabla_{pp}^2 Hp + O(h^3),
\end{aligned}
\tag{26}
$$

and

$$
\begin{aligned}
p(t_k + h) &= p + h\dot{p} + \tfrac{h^2}{2}\ddot{p} + O(h^3) \\
&= p - h\nabla_x H - h\gamma p + \tfrac{h^2}{2}\left(-\nabla_{xx}^2 H\dot{x} - \nabla_{xp}^2 H\dot{p} - \gamma\dot{p}\right) + O(h^3) \\
&= p - h\nabla_x H - h\gamma p - \tfrac{h^2}{2}\nabla_{xx}^2 H\nabla_p H + \tfrac{h^2}{2}\nabla_{xp}^2 H\nabla_x H + \tfrac{h^2}{2}\gamma\nabla_{xx}^2 Hp \\
&\quad + \tfrac{h^2}{2}\gamma\nabla_x H + \tfrac{h^2}{2}\gamma^2 p + O(h^3),
\end{aligned}
\tag{27}
$$

where we denote $x \equiv x(t_k)$ and $p \equiv p(t_k)$ for $t_k = kh$ ($k = 0, 1, \dots$), and it is implicit that all gradients and Hessians of $H$ are being computed at $(x, p)$.

Consider (10). Under one step of this map, starting from the point $(x, p)$, upon using Taylor expansions we have

$$
x_{k+1} = x + h\nabla_p H + O(h^2)
\tag{28}
$$

and

$$
p_{k+1} = e^{-\gamma h}p - h\nabla_x H + O(h^2) = p - \gamma h p - h\nabla_x H(x, p) + O(h^2).
\tag{29}
$$

Comparing these last two equations with (26) and (27) we conclude that

$$
x_{k+1} = x(t_k + h) + O(h^2), \qquad p_{k+1} = p(t_k + h) + O(h^2).
\tag{30}
$$

Therefore, the discrete state approximates the continuous state up to an error of $O(h^2)$, obeying Definition 1 with $r = 1$.

The same approach is applicable to the numerical map (11). Expanding the first update:

$$
\begin{aligned}
\tilde{X} &= x + \tfrac{h}{2}\nabla_p H\left(x + \tfrac{h}{2}\nabla_p H, p - \tfrac{h}{2}\gamma p\right) + O(h^3), \\
&= x + \tfrac{h}{2}\nabla_p H + \tfrac{h^2}{4}\nabla_{xp}^2 H\nabla_p H - \tfrac{h^2}{4}\gamma\nabla_{pp}^2 Hp + O(h^3).
\end{aligned}
\tag{31}
$$

Expanding the second update:

$$
\begin{aligned}
\tilde{P} &= e^{-\gamma h/2}p - \tfrac{h}{2}\nabla_x H\left(x + \tfrac{h}{2}\nabla_p H, p - \tfrac{h}{2}\gamma p\right) \\
&\quad - \tfrac{h}{2}\nabla_x H\left(x + \tfrac{h}{2}\nabla_p H, p - \tfrac{h}{2}\gamma p - h\nabla_x H\right) + O(h^3), \\
&= e^{-\gamma h/2}p - h\nabla_x H - \tfrac{h^2}{2}\nabla_{xx}^2 H\nabla_p H + \tfrac{h^2}{2}\gamma\nabla_{xp}^2 Hp + \tfrac{h^2}{2}\nabla_{xp}^2 H\nabla_x H + O(h^3).
\end{aligned}
\tag{32}
$$

Making use of (31) and (32) we thus find:

$$
\begin{aligned}
X &= \tilde{X} + \tfrac{h}{2}\nabla_p H(\tilde{X}, \tilde{P}) \\
&= x + \tfrac{h}{2}\nabla_p H + \tfrac{h^2}{4}\nabla^2_{xp}H\nabla_p H - \tfrac{h^2}{4}\gamma\nabla^2_{pp}Hp \\
&\quad + \tfrac{h}{2}\nabla H\big(x + \tfrac{h}{2}\nabla_p H, p - \tfrac{h}{2}\gamma p - h\nabla_x H\big) + O(h^3) \\
&= x + h\nabla_p H + \tfrac{h^2}{2}\nabla^2_{xp}H\nabla_p H - \tfrac{h^2}{2}\gamma\nabla^2_{pp}Hp - \tfrac{h^2}{2}\nabla^2_{pp}H\nabla_x H + O(h^3).
\end{aligned}
\tag{33}
$$

Comparing with (26) we conclude that

$$
x_{k+1} = x(t_k + h) + O(h^3). \tag{34}
$$

Finally, from (32) we have

$$
\begin{aligned}
P &= e^{-\gamma h/2}\tilde{P} \\
&= e^{\gamma h}p - e^{-\gamma h/2}\Big\{h\nabla_x H + \tfrac{h^2}{2}\nabla^2_{xx}H\nabla_p H + \tfrac{h^2}{2}\gamma\nabla^2_{xp}Hp - \tfrac{h^2}{2}\nabla^2_{xp}H\nabla_x H\Big\} + O(h^3) \\
&= p - \gamma h p + \tfrac{h^2}{2}\gamma^2 p - h\nabla_x H - \tfrac{h^2}{2}\nabla^2_{xx}H\nabla_p H + \tfrac{h^2}{2}\gamma\nabla^2_{xp}Hp \\
&\quad + \tfrac{h^2}{2}\nabla^2_{xp}H\nabla_x H + \tfrac{h^2}{2}\gamma\nabla_x H + O(h^3).
\end{aligned}
\tag{35}
$$

Comparing this with (27) implies

$$
p_{k+1} = p(t_k + h) + O(h^3). \tag{36}
$$

Therefore, in this case we satisfy Definition 1 with $r = 2$. $\qquad\square$

From the above general results it is immediate that:

- CM (1)—or equivalently (13) which is more appropriate to make connections with the continuous system—is a first order integrator to the conformal Hamiltonian system (3) with the classical Hamiltonian (4); the equations of motion are explicitly given by (38).
- The relativistic extension of CM given by (23) is a first order integrator to the conformal relativistic Hamiltonian system (22).
- RGD (24) with $\alpha = 1$—also equivalently written as Algorithm. 1—is a second order integrator to system (22).

## B   Insights into Nesterov and Heavy Ball Methods

Here we prove Theorem 6, but additionally provide several other details which give insights into Nesterov's method (NAG) and heavy ball or classical momentum (CM), such as their underlying "modified equations" and "shadow Hamiltonians."

### B.1   Order of Accuracy

We work on phase space variables $(x, p)$ thus NAG should be considered in the form (18), which we repeat below for convenience:

$$
x_{k+1/2} = x_k + \tfrac{h}{m}e^{-\gamma h}p_k, \tag{37a}
$$

$$
p_{k+1} = e^{-\gamma h}p_k - h\nabla f(x_{k+1/2}), \tag{37b}
$$

$$
x_{k+1} = x_k + \tfrac{h}{m}p_{k+1}. \tag{37c}
$$

Recall that this is precisely (2) under the change of variables (14). Let us now derive the order of accuracy of this method with respect to its underlying continuous Hamiltonian system:

$$
\dot{x} = p/m, \qquad \dot{p} = -\nabla f(x) - \gamma p. \tag{38}
$$

*Proof of Theorem 6: part (i).* Denoting $x = x(t_k)$ and $p = p(t_k)$, we expand the exponential in (37a) to obtain

$$
x_{k+1/2} = x + \tfrac{h}{m}p - \tfrac{h^2}{m}\gamma p + O(h^3). \tag{39}
$$

Using this and Taylor expansions in the last two updates (37b) and (37c) yield

$$p_{k+1} = p - h\gamma p - h\nabla f(x) + \frac{h^2}{2}\gamma^2 p - \frac{h^2}{m}\nabla^2 f(x)p_k + O(h^3), \tag{40a}$$

$$x_{k+1} = x + \frac{h}{m}p - \frac{h^2}{m}\gamma p - \frac{h^2}{m}\nabla f(x) + O(h^3). \tag{40b}$$

It is implicit that $\nabla f$ and $\nabla^2 f$ are computed at $(x, p)$. From the equations of motion (38), i.e. replacing the Hamiltonian (4) into the general approximations (26) and (27), we obtain

$$p(t_k + h) = p - h\nabla f - h\gamma p - \frac{h^2}{2m}\nabla^2 f p + \frac{h^2}{2}\gamma\nabla^2 f p + \frac{h^2}{2}\gamma\nabla f + \frac{h^2}{2}\gamma^2 p + O(h^3), \tag{41a}$$

$$x(t_k + h) = x + \frac{h}{m}p - \frac{h^2}{2m}\gamma p + O(h^3). \tag{41b}$$

Hence, by comparison with (40), we have $x_{k+1} = x(t_k+h)+O(h^2)$ and $p_{k+1} = p(t_k+h)+O(h^2)$, which according to Definition 1 means that NAG is an integrator of order $r = 1$, as claimed.  □

Both NAG and CM are first order integrators to the same continuous-time system (38). We already know that CM is conformal symplectic. Next we investigate the how NAG deforms the symplectic structure.

## B.2   Spurious Contraction of the Symplectic Form

*Proof of Theorem 6, part (ii).* Consider the variational form of (37) (the notation is standard in numerical analysis [30]):

$$dx_{k+1/2} = dx_k + \frac{h}{m}e^{-\gamma h}dp_k, \tag{42a}$$

$$dp_{k+1} = e^{-\gamma h}dp_k - h\nabla^2 f(x_{k+1/2})dx_{k+1/2}, \tag{42b}$$

$$dx_{k+1} = dx_k + \frac{h}{m}dp_{k+1}. \tag{42c}$$

Using these, bilinearity and the antisymmety of the wedge product, together the fact that $\nabla^2 f$ is symmetric, we obtain

$$\begin{aligned} dx_{k+1} \wedge dp_{k+1} &= dx_k \wedge dp_{k+1} \\ &= e^{-\gamma h}dx_k \wedge dp_k - h dx_k \wedge \nabla^2 f(x_{k+1/2})dx_{k+1/2} \\ &= e^{-\gamma h}dx_k \wedge dp_k - \frac{h^2}{m}e^{-\gamma h}dx_k \wedge \nabla^2 f(x_{k+1/2})dp_k \\ &= e^{-\gamma h}dx_k \wedge dp_k - \frac{h^2}{m}e^{-\gamma h}dx_k \wedge \nabla^2 f(x_k)dp_k + O(h^3), \end{aligned} \tag{43}$$

where in the last passage we used a Taylor approximation for $x_{k+1/2}$. Thus, $dx_{k+1} \wedge dp_{k+1} \neq e^{-\gamma h}dx_k \wedge dp_k$, showing that the method is not conformal symplectic (see Definition 2). Moreover, using the symmetry of $\nabla^2 f$ we can write (43) as

$$\omega_{k+1} = e^{-\gamma h}\left[I - \frac{h^2}{m}\nabla^2 f(x_k)\right]\omega_k + O(h^3), \tag{44}$$

which is exactly (19).  □

Thus, while CM exactly preserve the same dissipation found in the continuous-time system, NAG introduces some extra contraction or expansion of the symplectic form, depending whether $\nabla^2 f$ is positive definite or not. From (44), in $k$ iterations of NAG, and neglecting the $O(h^3)$ error term, we have

$$\begin{aligned} \omega_k &\approx e^{-\gamma t_k}\prod_{i=1}^{k}\left[I - \frac{h^2}{m}\nabla^2 f(x_{k-i})\right]\omega_0 \\ &\approx e^{-\gamma t_k}\left[I - \frac{h^2}{m}\big(\nabla^2 f(x_{k-1}) - \nabla^2 f(x_{k-2}) - \cdots - \nabla^2 f(x_0)\big)\right]\omega_0. \end{aligned} \tag{45}$$

This depends on the entire history of the Hessians from the initial point. Therefore, NAG contracts the symplectic form slightly more than the underlying conformal Hamiltonian system—assuming $\nabla^2 f$ is positive definite—and it does so in a way that depends on the Hessian of the objective function. Note that this is a small effect of $O(h^2)$. Moreover, if $\nabla^2 f$ has negative eigenvalues, e.g.

$f$ is nonconvex and has saddle points, then NAG actually introduces some spurious excitation in that direction. To gain some intuition, let us consider the simple case of a quadratic function:[8]

$$f(x) = (\lambda/2)x^2 \tag{46}$$

for some constant $\lambda$. Thus (44) becomes

$$\omega_{k+1} \approx e^{-\gamma h + \log(1 - h^2 \lambda/m)} \omega_k \approx e^{-(\gamma + h\lambda/m)h} \omega_k \implies \omega_k \approx e^{-(\gamma + h\lambda/m)t_k} \omega_0. \tag{47}$$

This suggests that effectively the original damping of the system is being replaced by $\gamma \to \gamma + h\lambda/m$. Thus if $\lambda > 0$ there is some spurious damping, whereas if $\lambda < 0$ there is some spurious excitation.

### B.3 Modified Equations and Shadow Hamiltonian

We have seen above that NAG is a first order integrator to the conformal Hamiltonian system (38), however it changes slightly the behaviour of the original system since it introduces spurious damping or excitation. To understand its behaviour more closely, one can ask the following question: *for which continuous dynamical system, NAG turns out to be a second order integrator?* In other words, we can look for a modified system that captures the behaviour of NAG more closely, up to $O(h^3)$. Every numerical method is known to have a modified or perturbed differential equation [30] (the brief discussion in [18] may also be useful). In answering this question, we thus find the following.

**Theorem 7** (Shadow dynamical system for Nesterov's method). *NAG* (2)*, or its equivalent phase space representation* (37)*, is a second order integrator to the following modified or perturbed equations:*

$$\dot{x} = \frac{1}{m}p - \frac{\gamma h}{2m}p - \frac{h}{2m}\nabla f(x), \qquad \dot{p} = -\nabla f(x) - \gamma p - \frac{h\gamma}{2}\nabla f - \frac{h}{2m}\nabla^2 f(x)p. \tag{48}$$

*Proof.* We look for vector fields $F(q, p; h)$ and $G(q, p; h)$ for the modified system

$$\dot{x} = p/m + hF, \qquad \dot{p} = -\nabla f(x) - \gamma p + hG, \tag{49}$$

such that (37) is an integrator of order $r = 2$. This can be done by computing [30]

$$F = \lim_{h \to 0} \frac{x_{k+1} - x(t_k + h)}{h^2}, \qquad G = \lim_{h \to 0} \frac{p_{k+1} - p(t_k + h)}{h^2}. \tag{50}$$

From (40) and (41) we obtain precisely (48). By the previously discussed approach through Taylor expansions one can also readily check that NAG is indeed an integrator of order $r = 2$ to this perturbed system. $\qquad \square$

We can also combine (48) into a second order differential equation:

$$m\ddot{x} + m\left(\gamma I + \frac{h}{m}\nabla^2 f(x)\right)\dot{x} = -\left(I + \frac{h\gamma}{2}I - \frac{h^2\gamma^2}{4}I + \frac{h^2}{4m}\nabla^2 f(x)\right)\nabla f(x), \tag{51}$$

where $I$ is the $n \times n$ identity matrix. We see that this equation has several new ingredients compared to

$$\ddot{x} + \gamma\dot{x} = -(1/m)\nabla f(x), \tag{52}$$

which is equivalent to (38). First, when $h \to 0$ the system (51) recovers (52), as it should since both must agree to leading order. Second, the spurious change in the damping coefficient reflects the behaviour of the symplectic form (44) (see also (47)). Third, we see that the gradient $\nabla f$ is rescaled by the contribution of several terms, including the Hessian $\nabla^2 f$, making explicit a curvature dependent behaviour, which also appears in the damping coefficient. Note that the modified equation (51), or equivalently (48), depends on the step size $h$, hence it captures an intrinsic behaviour of the discrete-time algorithm that is not captured by (38).

Since CM is also a first order integrator to (38), which is actually conformal symplectic, it is natural to consider its modified equation and compare with the one for NAG (48). We thus obtain the following.

**Theorem 8** (Shadow Hamiltonian for Heavy Ball). *The heavy ball or CM method* (1), *equivalently written in phase space as* (13)*, is a second order integrator to the following modified conformal Hamiltonian system:*

$$\dot{x} = \frac{1}{m}p - \frac{h\gamma}{2m}p - \frac{h}{2m}\nabla f(x), \qquad \dot{p} = -\nabla f(x) - \gamma p - \frac{h\gamma}{2}\nabla f(x) + \frac{h}{2m}\nabla^2 f(x)p. \quad (53)$$

*Such a system admits the shadow or perturbed Hamiltonian*

$$\tilde{H} = \frac{1}{2m}\|p\|^2 + f(x) - \frac{h\gamma}{4m}\|p\|^2 - \frac{h}{2m}\langle\nabla f(x), p\rangle + \frac{h\gamma}{2}f. \quad (54)$$

*Proof.* It follows exactly as in Theorem 7. Also, one can readily verify that replacing (54) into (3) gives (54). □

We note the striking similarity between (53) and (48); the only difference is the sign of the last term in the second equation. Up to this level of resolution, the difference is that NAG introduces a spurious damping compared to CM, in agreement with the derivation of the symplectic form (44). On the other hand, notice that the perturbed system (53) for CM is conformal Hamiltonian, contrary to (48) that cannot be written in Hamiltonian form; this is the reason why structure-preserving discretizations tend to be more stable since the perturbed trajectories are always close, i.e. within a bounded error, from the original Hamiltonian dynamics. We can also combine (53) into

$$m\ddot{x} + m\gamma\dot{x} = -\left(I + \frac{h\gamma}{2}I - \frac{h^2\gamma^2}{4}I - \frac{h^2}{4m}\nabla^2 f(x)\right)\nabla f(x). \quad (55)$$

Again, this is strikingly similar to (51). Note that this equation does not have the spurious damping term $(h/m)\nabla^2 f(x)$ as in (51), making even more explicit that it preserves exactly the dissipation of the original continuous system. As we will show below, there is a balance between preserving such a dissipation and the stability of the method. While NAG introduces an extra damping, and may slightly help in an improved convergence since it dissipates more energy, this comes at the price in a decreased stability.

## C   Tradeoff Between Stability and Convergence Rate

Here we illustrate an interesting phenomenon: there is a tradeoff between stability versus convergence rate. Intuitively, an improved rate is associated to a higher "contraction," i.e. the introduction of spurious dissipation in the numerical method. However, this makes the method less stable, and ultimately very sensitive to parameter tuning. On the other hand, a geometric or structure-preserving integrator may have slightly less contraction, since it preserves the original dissipation of the continuous system exactly, but it is more stable and able to operate with larger step sizes. Furthermore, a structure-preserving method is guaranteed to reproduce very closely, perhaps even up to a negligible error, the continuous-time rates of convergence [18]. This indicates that there may have benefits in considering this class of methods for optimization, such as conformal symplectic integrators that are being advocated in this paper.

Stability of a numerical integrator means the region of hyperparameters, e.g. values of the step size, such that the method is able to converge. The larger this region, more stable is the method. The convergence rate is a measure of how fast the method tends to the minimum, and this is related to the amount of contraction between subsequent states, or subsequent values of the objective function. For instance, since NAG introduces some spurious dissipation (recall (44)) we expect that it may have a slightly higher contraction compared to CM, which exactly preserves the dissipation of the continuous system. Thus, such a spurious dissipation can induce a slightly improved convergence rate, but as we will show below, at the cost of making the method more unstable and thus requiring smaller step sizes.

Let us consider a standard linear stability analysis, which involves a quadratic function (46) such that the previous methods can be treated analytically. Thus, replacing (46) into CM in the form (13) it is possible to write the algorithm as a linear system:

$$z_{k+1} = T_{CM}z_k, \qquad T_{CM} = \begin{bmatrix} 1 - h^2\lambda/m & (h/m)e^{-\gamma h} \\ -h\lambda & e^{-\gamma h} \end{bmatrix}, \quad (56)$$

Figure 3: Stability of CM (13) (*blue*), NAG (18) (*green*), and RGD (24) with $c \to \infty$ and $\alpha = 1$ (*black*)—in this case it becomes a dissipative version of the Leapfrog to system (38). We plot the eigenvalues in the complex plane; $x$-axis is the real part, $y$-axis is the imaginary part. The unit circle represent the stability region, i.e. once an eigenvalue leaves the gray area the corresponding method becomes unstable. Both CM and RGD are symplectic thus their eigenvalues always move on a circle of radius $e^{-\gamma h/2}$ centered at the origin. NAG has eigenvalues in the smaller circle with radius $1/(e^{\gamma h} + 1)$ and centered at $1/(e^{\gamma h} + 1)$ on the $x$-axis; the circle is dislocated from the origin precisely due to spurious dissipation. From left to right we increase the step size $h$ while keeping $\gamma$, $m$, and $\lambda$ fixed. As $h$ increases the eigenvalues move on the circles in the counterclockwise direction until they fall on the real line. Eventually they leave the unit circle and the associated method becomes unstable. Note how CM has higher stability than NAG, and RGD has even higher stability than CM.

where we denote $z = \begin{bmatrix} x \\ p \end{bmatrix}$. Similarly, NAG in the form (18) yields

$$z_{k+1} = T_{\text{NAG}} z_k, \qquad T_{\text{NAG}} = \begin{bmatrix} 1 - h^2\lambda/m & (h/m)e^{-\gamma h}(1 - h^2\lambda/m) \\ -h\lambda & e^{-\gamma h}(1 - h^2\lambda/m) \end{bmatrix}, \qquad (57)$$

while RGD (24), with $c \to \infty$ and $\alpha = 1$, yields[9]

$$z_{k+1} = T_{\text{RGD}} z_k, \qquad T_{\text{RGD}} = \begin{bmatrix} 1 - h^2\lambda/(2m) & h/(2m)e^{-\gamma h/2}(2 - h^2\lambda/(2m)) \\ -h\lambda e^{-\gamma h/2} & e^{-\gamma h}(1 - h^2\lambda/(2m)) \end{bmatrix}. \qquad (58)$$

A linear system is stable if the spectral radius of its transition matrix is $\rho(T) \leq 1$. We can compute the eigenvalues of the above matrices and check for which range of parameters they remain inside the unit circle; e.g. for given $\gamma$, $m$, and $\lambda$ we can find the allowed range of the step size $h$ for which the maximum eigenvalue in absolute value is $|\lambda_{\max}| \leq 1$. Instead of showing the explicit formulas for these eigenvalues, which can be obtained quite simply but are cumbersome, let us illustrate what happens graphically.

In Fig. 3, the shaded gray area represents the unit circle. Any eigenvalue that leaves this area makes the associated algorithm unstable. Here we fix $m = \lambda = \gamma = 1$ (other choices are equivalent) and we vary the step size $h > 0$. These eigenvalues are in general complex and lie on a circle which is determined by the amount of friction in the system. Note how for CM and RGD this circle is centered at the origin, with radius $\sqrt{\mu} \equiv e^{-\gamma h/2}$, since these methods are conformal symplectic and exactly preserve the dissipation of the underlying continuous system. However, NAG introduces a spurious damping which is reflected as the circle being translated from the center, at a distance $1/(e^{\gamma h} + 1)$, and moreover this circle has a smaller radius of $1/(e^{\gamma h} + 1)$ compared to CM and RGD; since this radius is smaller, NAG may have a faster convergence when these eigenvalues are complex. As we increase $h$ (left to right in Fig. 3), the eigenvalues move counterclockwise on the circles until falling on the real line, where one of them goes to the left while the other goes to the right. Eventually, the leftmost eigenvalue leaves the unit circle for a large enough $h$ (third panel in Fig. 3). Note that NAG becomes unstable first, followed by CM, and only then by RGD. The main point is that CM and RGD can still be stable for much larger step sizes compared to NAG, and RGD is even more stable than CM as seen in the rightmost plot in Fig. 3; this is a consequence of RGD being an integrator of order $r = 2$ whereas CM is of order $r = 1$. Hence, even though NAG may

have a slightly faster convergence (due to a stronger contraction), it requires a smaller step sizes and its stability is more sensitive compared to a conformal symplectic method. On the other hand, both CM and RGD can operate with larger step sizes, which in practice may even result in a faster solver compared to NAG.

To provide a more quantitative statement, after computing the eigenvalues of the above transition matrices for given $\mu \equiv e^{-\gamma h}$, $m$, and $\lambda$, we find the following threshold for stability:

$$h_{\text{CM}} \leq \sqrt{m(1 + \mu + \mu^2 + \mu^3)}/(\mu\sqrt{\lambda}), \tag{59}$$

$$h_{\text{NAG}} \leq \sqrt{m(1 + \mu + \mu^2 + \mu^3)}/\sqrt{\mu\lambda(1 + \mu + \mu^2)}, \tag{60}$$

$$h_{\text{RGD}} \leq \sqrt{2m(1 + \mu + \mu^2 + \mu^3)}/\sqrt{\mu\lambda(1 + \mu)}. \tag{61}$$

We can clearly see that RGD has the largest region for $h$, followed by CM, then by NAG, in agreement with the results of Fig. 3.

# D  Additional Numerical Experiments

Here we compare RGD (Algorithm 1) with CM (1) and NAG (2) on several additional test functions; for details on these functions see e.g. [39] and references therein. We follow the procedure already described in Section 6 where we optimized the hyperparameters of these algorithm using Bayesian optimization.[10] We report the convergence rate using the best parameters found together with histograms of the parameter search. In all cases we initialize the velocity as $v_0 = 0$. The initial position $x_0$ was chosen inside the range where the corresponding test function is usually considered.

First we consider functions with a quadratic growth. These results are shown in Figs. 4–7. In this case RGD performed similarly to CM and NAG, although with some improvement. In any case RGD proved to be more stable, i.e. it worked well for a wider range of hyperparameters.

We expect that RGD stands out on settings with large gradients or objective functions with fast growing tails. Therefore, in the remaining figures, i.e. Fig. 8–15, we consider more challenging optimization problems with functions that grow stronger than a quadratic. For some of these problems the minimum lies on a flat valley, making it hard for an algorithm to stop around the minimum after gaining a lot of speed from a very steep descent direction. Note that in all these cases the improvement of RGD over CM and NAG is significant, and the parameter $\delta$—which controls relativistic effects—had an important role. The conformal symplecticity, which is indicated by the tendency $\alpha \to 1$, also brings an improved stability in the discretization. These results provide compelling evidence for the benefits of RGD.

Figure 4: *Booth function*: $f(x, y) \equiv (x + 2y - 7)^2 + (2x + y - 5)^2$. Global minimum at $f(1, 3) = 0$. We initialize at $x_0 = (10, 10)$. This function is usually evaluated on the region $-10 \leq x, y \leq 10$. All methods perform well on this problem which is not challenging.

Figure 5: *Matyas function*: $f(x, y) \equiv 0.26(x^2 + y^2) - 0.48xy$. Global minimum is at $f(0, 0) = 0$. We initialize at $x_0 = (10, -7)$. This function is usually evaluated on the region $-10 \leq x, y \leq 10$. Even though the function has a—not so strong—quadratic growth, we see a slight improvement of RGD; note $\delta > 0$. Note also the "symplectic tendency" $\alpha \to 1$.

Figure 6: *Lévi function #13*: $f(x, y) \equiv \sin^2 3\pi x + (x - 1)^2(1 + \sin^2 3\pi y) + (y - 1)^2(1 + \sin^2 2\pi y)$. This function is multimodal, with the global minimum at $f(1, 1) = 0$. We initialize at $x_0 = (10, -10)$. This function is usually studied on the region $-10 \leq x, y \leq 10$. Although this function is nonconvex, the optimization problem is not very challenging. However, we noticed that CM and NAG got stuck on a local minimum more often than RGD when running this example multiple times.

Figure 7: *Sum of squares*: $f(x) \equiv \sum_{i=1}^{n} i x_i^2$. The minimum is at $f(0) = 0$. We consider $n = 100$ dimensions and initialize at $x_0 = (10, \ldots, 10)$. The usual region of study is $-10 \leq x_i \leq 10$. Note that there is a clear tendency towards $\alpha \to 1$ in this case, i.e. in being conformal symplectic.

Figure 8: *Beale function*: $f(x, y) \equiv (1.5 - x + xy)^2 + (2.25 - x + xy^2)^2 + (2.625 - x + xy^3)^2$. The global minimum is at $f(3, 1/2) = 0$, lying on a flat and narrow valley which makes optimization challenging. Note also that this functions grows stronger than a quadratic. This function is usually considered on the region $-4.5 \leq x, y \leq 4.5$. We initialize at $x_0 = (-3, -3)$. Note how CM and NAG were unable to minimize the function, while RGD was able to find the global minimum to high accuracy; $\delta \gg 0$ played a predominant role, indicating benefits from "relativistic effects."

Figure 9: *Chung-Reynolds function*: $f(x) \equiv \left( \sum_{i=1}^{n} x_i^2 \right)^2$. The global minimum is at $f(0) = 0$. This function is usually considered on the region $-100 \leq x_i \leq 100$. We consider $n = 50$ dimensions and initialize at $x_0 = (50, \ldots, 50)$. Note that RGD was able to improve convergence by controlling the kinetic energy with $\delta > 0$. We also see the benefits of being conformal symplectic, i.e. $\alpha \to 1$.

Figure 10: *Quartic function*: $f(x) \equiv \sum_{i=1}^{n} i x_i^4$. The global minimum is at $f(0) = 0$. This function is usually considered over $-1.28 \leq x_i \leq 1.28$. We choose $n = 50$ dimensions and initialize at $x_0 = (2, \ldots, 2)$.

Figure 11: *Schwefel function*: $f(x) \equiv \sum_{i=1}^{n} x_i^{10}$. The minimum is at $f(0) = 0$. The function is usually considered over $-10 \le x_i \le 10$. This function grows even stronger than the previous two cases. We consider $n = 20$ dimensions and initialize at $x_0 = (2, \ldots, 2)$. Note that $\delta > 0$ is essential to control the kinetic energy and improve convergence.

Figure 12: *Qing function*: $f(x) \equiv \sum_{i=1}^{n} (x_i^2 - i)^2$. This function is multimodal, with minimum at $x_i^\star = \pm\sqrt{i}$, $f(x^\star) = 0$. The function is usually studied in the region $-500 \le x_i \le 500$. We consider $n = 100$ dimensions with initialization at $x_0 = (50, \ldots, 50)$.

Figure 13: *Zakharov function*: $f(x) \equiv \sum_{i=1}^{n} x_i^2 + \left(\frac{1}{2}\sum_{i=1}^{n} i x_i\right)^2 + \left(\frac{1}{2}\sum_{i=1}^{n} i x_i\right)^4$. The minimum is at $f(0) = 0$. The region of interest is usually $-5 \le x_i \le 10$. We consider $n = 5$ and initialize at $x_0 = (1, \ldots, 1)$. Note that $\delta > 0$ played a dominant role here, and $\alpha \to 1$ as well. RGD successfully minimized this function to high accuracy, contrary to CM and NAG that were unable to get even close to the minimum.

Figure 14: Three-hump camel back function: $f(x,y) \equiv 2x^2 - 1.05x^4 + x^6/6 + xy + y^2$. This is a multimodal function with global minimum is at $f(0,0) = 0$. The region of interest is usually $-5 \leq x, y \leq 5$. We initialize at $x_0 = (5,5)$. The two local minima are somewhat close to the global minimum which makes optimization challenging. Only RGD was able to minimize the function.

Figure 15: *Rosenbrock function*: $f(x) \equiv \sum_{i-1}^{n-1} \left( 100(x_{i+1} - x_i^2)^2 + (x_i - 1)^2 \right)$. The global minimum is at $f(1, \ldots, 1) = 0$. More details about this function was described in Section 6. Here we consider $n = 1000$ dimensions and initialize at $x_0 = (2.048, \ldots, 2.048)$. This function is usually studied in the region $-2.048 \leq x_i \leq 2.048$. Note that $\delta > 0$ was important for the improved convergence.

## Footnotes

[8] This quadratic function is actually enough to capture the behaviour when close to a critical point $x^\star$ since $f(x) \approx f(x^\star) + \frac{1}{2}\nabla^2 f(x^\star)x$ and one can work on rotated coordinates where $\nabla^2 f(x^\star) = \mathrm{diag}(\lambda_1, \ldots, \lambda_n)$.

[9]The case of finite $c$ is nonlinear and not amenable to such an analysis. However, this $c \to \infty$ already provide useful insights.

[10]We provide the actual code used in our numerical simulations in [34].