[Reviews · NeurIPS 2020]

Review 1

Summary and Contributions: The paper investigates optimization algorithms through the lense of dynamical systems. This connection allows to transfer known results from dynamical systems to optimization. In particular, the paper investigates two known algorithms (heavy ball and Nesterov's accelerated descent) and proposes a new one coined "relativistic gradient descent". The paper investigates these algorithms in terms of order approximation, conformal simplecticity and dissipation. In addition, 4 benchmark examples show that the new proposed algorithm converges faster than the other two algorithms, even in the case when all parameters have been tuned "optimally".

Strengths: The paper is interesting, largely sound, a novel contribution to the literature and of relevance to the NeurIPS community. Based on the presented information, I believe the theoretical results are correct. The numerical results are interesting since they cover a range of relevant problems and fair since all parameters have been tuned "optimally" using Bayesian optimization. The connection of optimization with dynamical systems is very relevant to the NeurIPS community since understanding and enhancing optimization is fundamentally important. The proposed algorithm is novel.

Weaknesses: Although the paper has a number of strengths, it also has some weaknesses. While the theoretical results of the paper are sound and important from a dynamical systems perspective, it remains questionable how important these are in an optimization context. In any case the authors fail to convince me in this regard. For the numerical results, all parameters were tuned "optimally" using Bayesian optimization. This is a good and fair approach. However, the details here are vague and it is not clear for instance what the metric was under which the parameters have been tuned. That being said, I trust that the authors chose a relevant metric.

Correctness: The claims in the paper are largely correct. One exception is for instance the claim in page 2 that "RGD never diverges". This claim is based on the fact that the difference of certain iterates stay globally bounded. There is absolutely no reason why such conclusion should be true. In fact, the authors themselves observed on page 7 that "CM and NAG diverged more often than RGD" implying that RGD did diverge in a couple of their numerical examples.

Clarity: Yes, the paper is extremely well written. It is well structured and can be easily digested.

Relation to Prior Work: Yes, prior work is appropriately referenced.

Reproducibility: Yes

Additional Feedback: After reading the other reviews, the authors's feedback and discussion with other reviewers, I have decided to change my score to "8" and have updated my review accordingly. My main concern has been clearly resolved.


Review 2

Summary and Contributions: The authors propose to apply the conformal symplectic method to compute the Nesterov’s accelerated gradient and Polyaks’s heavy ball. In particular, they study the possibility of realistic damped Hamiltonian flow. They propose a new algorithm that normalizes the momentum and may result in more stable/faster optimization Here the realistic Hamiltonian helps in the simple update of momentum with a little additional computational cost.

Strengths: The accelerated gradient flow is important in optimization. The paper addresses the problem in using damped Hamiltonian flows with conformal symplectic method.

Weaknesses: 1. This work study several toy examples in purely optimization problems. It may be interesting to study machine learning optimization problems, or Bayesian sampling problem. 2. Is there a theoretical justification that shows the realistic Hamitlonian is better than Euclidean Hamiltonian? Does it provide a new convexity which helps the optimization convergence for some particular functions or non-convex functions?

Correctness: The claim is correct by using the damped Hamiltonian flow with conformal Symplectic discretization.

Clarity: The paper is well written.

Relation to Prior Work: There are also related work on applying damped Hamiltonian flows in probability density space, which works for Bayesian sampling. Yifei Wang, Wuchen Li. Accelerated Information Gradient flow, arXiv:1909.02102.

Reproducibility: Yes

Additional Feedback: For numerical expreiments, more machine learning examples may be needed. In addition, a toy example which explains why the proposed method works may be useful. It could explain why the realistic Hamiltonian is better than the classical quadratic kinetic energy in these problems. I have read the authors' response. They carefully address my questions. This is a good attempt in the generalization of Nesterov algorithm, which could have border applications in sampling. I increase my score from 6 to 7.


Review 3

Summary and Contributions: This paper brings ideas from the physics literature to study and development of gradient-based optimization methods. By considering specific structure-preserving discretization of continuous time dynamical systems, the authors shed a new light on the theoretical grounding of classical momentum (CM) and Nesterov accelerated gradient (NAG). Moreover, they derive a new interesting optimization algorithm called relativistic gradient descent (RGD) interpolating the behavior of CM and NAG. Once properly tuned, the authors show empirical evidence of the better performance of their method over CM and NAG.

Strengths: This work sheds a new light on the theoretical grounding of CM and NAG. A new interesting optimization algorithm RGD is proposed. It interpolates between the structural properties of both CM and NAG. Crucial structural and theoretical properties of RGD along with CM and NAG are derived and certainly provide a better understanding of the methods. The experimental section clearly supports the theoretical claims. I am a strong advocate of the claim l319 "cross-fertilization between dynamical systems theory, physics, machine learning and optimization". The pedagogical efforts made by the authors is much appreciated.

Weaknesses: There are 2 extra hyper-parameters to tune compared to CM and NAG. It would have been great to be able to play with some supplementary code, please consider releasing the corresponding implementation.

Correctness: I cannot judge the correctness of the claims and proofs. The empirical methodology seems clean, parameter tuning is performed via Bayesian optimization and is well interpreted. Could you give the computational infrastructure and programming language which allows you to monitor 10^-34 variations?

Clarity: The paper is very well written, it is a pedagogical work to bring up physical jargon to the ML audience.

Relation to Prior Work: Yes it is. Nevertheless, I don't understand why this work is listed in the Deep Learning track and not the Optimization one.

Reproducibility: Yes

Additional Feedback: l 136 t_k notation is only implicitly defined Figure 3 in appendix might be used in the main text to illustrate the stability properties of each procedures and to make the body less dense. At least, l 69 add reference to Figure 3 Please make sure the hyperlinks associated to citations, equations, etc. are clickable in the main text. # EDIT: The authors' response and the discussion with the other reviewers have convinced me to support even more this contribution despite my low expertise on the topic.

[Author Response · NeurIPS 2020]

**R1:** We appreciate the reviewer's positive comments and for recognizing the novelty and importance of our work.

• "a similar algorithm already exists in the literature which the authors fail to cite nor to compare against" As per

NeurIPS reviewing instructions: *"authors are excused for not knowing about all non-refereed work (e.g, those appearing*

*on arXiv) . . ."* This already anticipates that extra care is in order when comparing to arXiv to avoid misunderstanding.

We are quite aware of this paper and reiterate: our paper is original and the first with such an approach.

• "While theoretical results are sound and important from a dynamical systems perspective, it remains questionable

in an optimization context." The significance of our work to optimization is that it opens the door for building new

optimization algorithms from first principles by applying a general structure-preserving discretization scheme, i.e.

Eq. (12), to a dynamical system. Another reason why our approach is relevant to optimization can be found in *"On*

*Dissipative Symplectic Integration with Applications to Gradient-Based Optimization"* [arxiv.org/abs/2004.06840],

Ref. [18]; see also L182–193 in the paper where it is shown that structure-preserving discretizations allow one to

preserve rates of convergence. Symplectic discretizations of dissipative systems ensure that their "good-properties"

are transferred automatically to discrete-time; we also provided compelling evidence why the relativistic system is

relevant for optimization (L225–239, numerics, stability in the supplement, etc.). We will clarify and provide more

intuition in the revision. We mention in passing that formal convergence-rate-results for the relativistic system are very

challenging and beyond scope—see L307–314—however such a method approaches CM or NAG as limiting cases,

hence its convergence cannot be worse than CM or NAG, which are well-understood in the literature.

• "The details of the numerics are vague . . . relevant metric." The implementation is simple, exactly as written, i.e.,

Algorithm 1, Eq. (1) and Eq. (2). The Bayesian optimization follows the original Ref. [34] (we used the Python library

"hyperopt" provided by the authors). The parameters were optimized to give the smallest possible objective function

value. We agree that we could have provided more details. Since space is short, we will include more comments about

this in the supplement. Moreover, we will make our code publicly available. Thank you for bringing this up.

• "RGD never diverges." This was a misleading "abuse of language" on our part and will be corrected. Thank you.

**R3:** We would like to thank the reviewer for comments and suggestions, which will be incorporated.

• *Weaknesses/additional feedback:* "More machine learning problems in the numerical section. A toy example . . ."

We agree that practical ML problems are worth exploring. We show some preliminary results in the Figures below. We

will improve these and include more ML experiments in the supplement (and also available code). In passing, let us

mention that leading experts in the field have emphasized the great need for theoretical and principled approaches to

construct new algorithms, as well as understand existing ones. *Our contributions are on these theoretical lines.* We

hope the reviewer may appreciate their value, independent of practical experiments (although we provided some that

confirm our claims). Our paper brings, and extends, important ideas from physics to ML/optimization.

Regarding the toy problem, the Rosenbrock case in the paper is a meaningful example. *We expect that our method,*

*RGD, stands out when the objective has fast growing tails*; since RGD can control the "velocity" without having to

reduce the step size, it can navigate through such landscapes more effectively. We will clarify and provide a couple of

intuitive examples in the revision, such as in Fig. (e) below for a one-dimensional case. Thanks for raising this question.

"Is there a theoretical justification that shows the relativistic Hamiltonian is better than Euclidean . . ." This is a very deep

question, whose answer requires some serious differential geometry; we just started to understand this and unfortunately

is beyond scope. In short, a Hamiltonian dynamics happens in the cotangent bundle, $T^*\mathcal{M}$, of the base manifold $\mathcal{M}$

where the objective $f$ is defined. The Lagrangian formalism happens on the tangent bundle $T\mathcal{M}$. The kinetic energy

defines a metric on $T\mathcal{M}$ yielding different "momentum lengths." Thus different kinetic energies may yield better

algorithms if they can "match/adapt" to the geometry of $f$; the relativistic is just one example. We believe there is a

deep relation (i.e. an equation) that relates dynamical quantities to the duality gap, which may help to elucidate this.

• *Prior work:* Thank you for pointing out "Wang & Li (2019)," we will include a citation in the revised version.

Left to right. CM always close to NAG (when not shown). **RGD=ours**. (a) Movie Lens; SoftImput=[Hastie et al JMLR (2015)]. (b) Matrix Completion with noise; Error $10^{-5}$ of oracle bound [Candes&Plan IEEE (2009)]. (c) MNIST, feed forward CNN (3 layers), cross entropy. (d) VGG net (11 layers). (e) $f(x) = (1/8)(x^2 + 1)^4 - 1/8$; RGD stands out on fast growing tails.

**R4:** We appreciate R4's clear summary of our work. His/her understanding is very accurate, so we are surprised by

his/her confidence score. We will certainly incorporate the "additional feedback." We are also encouraged to hear that

the reviewer supports this line of work at the interface of dynamical systems, physics, and ML.

• "There are 2 extra hyperparameters compared to CM and NAG." (There is no free lunch!) In practice, there is

actually only 1 extra parameter. We included $\alpha$ in Algo. 1 only to illustrate, in an unbiased manner, that preserving the

symplectic structure can indeed be beneficial; all results in Fig. 2 favor $\alpha \to 1$, as opposed to $\alpha \to 0$. Thus, one should

fix $\alpha = 1$. We will clarify this in the revision. Indeed, we will provide publicly available code.

• "Computational infrastructure and programming language" We used Python 3 on a Mac Book Pro, Quad-Core Intel

i5, 16 GB RAM, and standard libraries such as numpy, scipy, hyperopt.

[Meta-Review · NeurIPS 2020]

A *very* original and nice paper that the reviewers clearly enjoyed. I would like to personally thank the authors for an inspiring work.